# Insulin Resistance and Diabetes Mellitus in Alzheimer’s Disease

**DOI:** 10.3390/cells10051236

**Published:** 2021-05-18

**Authors:** Jesús Burillo, Patricia Marqués, Beatriz Jiménez, Carlos González-Blanco, Manuel Benito, Carlos Guillén

**Affiliations:** 1Department of Biochemistry, Complutense University, 28040 Madrid, Spain; jburillo@ucm.es (J.B.); pmarques@ucm.es (P.M.); bejime02@ucm.es (B.J.); carlgo23@ucm.es (C.G.-B.); mbenito@ucm.es (M.B.); 2Centro de Investigación Biomédica en Red (CIBER) de Diabetes y Enfermedades Metabólicas Asociadas (CIBERDEM), 28040 Madrid, Spain; 3Mechanisms of Insulin Resistance (MOIR2), General Direction of Universities and Investigation (CCMM), 28040 Madrid, Spain

**Keywords:** insulin resistance, T3DM, mTOR, ER stress, autophagy, inflammation

## Abstract

Type 2 diabetes mellitus is a progressive disease that is characterized by the appearance of insulin resistance. The term insulin resistance is very wide and could affect different proteins involved in insulin signaling, as well as other mechanisms. In this review, we have analyzed the main molecular mechanisms that could be involved in the connection between type 2 diabetes and neurodegeneration, in general, and more specifically with the appearance of Alzheimer’s disease. We have studied, in more detail, the different processes involved, such as inflammation, endoplasmic reticulum stress, autophagy, and mitochondrial dysfunction.

## 1. Introduction

Diabetes is a metabolic disease that is characterized by the appearance of chronic hyperglycemia because of pancreatic β cell failure by different mechanisms. This decline of β cells occurs in all types of diabetes, but it is an essential mechanism in the main forms of diabetes; type 1 diabetes mellitus (T1DM) and type 2 diabetes mellitus (T2DM) [1]. According to American Diabetes Association (ADA), diabetes can be classified into different categories: Type 1 diabetes mellitus (T1DM); type 2 diabetes mellitus (T2DM); gestational diabetes mellitus (GDM); other causes of the disease, including monogenic diabetes, such as maturity-onset diabetes of the young (MODY), or secondary to the use of different drugs or chemical compounds [2].

T1DM is known as “insulin dependent” and represents 5–10% of the total amount of diabetics. The etiology of the disease is an autoimmune attack towards pancreatic β cells. Although there are several genetic predispositions related to the disease [3,4], it is also considered the existence of an environmental component, which is poorly understood [5,6].

T2DM is known as “insulin independent” and represents the majority of all diabetics (90–95%). The main characteristic of the disease is the appearance of insulin resistance, which is a defect in insulin signaling and a correct coupling of insulin with its receptor. It can be distinguished 2 phases in the disease progression. During the first one, there is insulin resistance, and concomitantly, a compensatory mechanism in pancreatic β cells, associated with hyperinsulinemia [7,8]. Although T2DM is considered non-insulin dependent, it is known that as the disease progresses, many patients need insulin because of pancreatic β cell destruction. Then, β cell function maintenance is a key treatment strategy [9]. Although β cell death is one of the main events of pancreatic β cell failure, some authors indicate that dedifferentiation is another important mechanism for β cell dysfunction [10]. In this regard, it has been observed that β cells can dedifferentiate to α and δ cells in human diabetic donors [11].

GDM is the appearance of insulin resistance during pregnancy and its maintenance after delivery. During gestation, all women suffer a series of both physical and metabolic changes, which make them more susceptible to develop insulin resistance [12]. Since GDM facilitates the development of T2DM after delivery, women with a GDM diagnose should be monitored for prediabetes and T2DM.

Monogenic diabetes are a heterogeneous group of diseases, which are the most common group of disorders affecting pancreatic β cells by a mutation in one gene. Represents 1–5% of all diabetics, and the majority of these mutations alter insulin secretion. There are 14 different variants of MODY (For instance, there are mutations that affect glucokinase (GK) gene, and it is known as MODY2, or different transcription factors which are involved in the maturation of pancreatic β cells, such as pancreatic and duodenal homeobox 1 (Pdx1), known as MODY 4). Apart from Pdx1 and GK, there is another important group of mutated genes that is involved in MODY and are the hepatocyte nuclear factors (HNFs) [13,14,15].

Once insulin is recognized by insulin receptor (IR), there is a conformational change in its structure, and the activation of the endogenous tyrosine kinase activity of β subunits. Phosphotyrosine residues permit the recruitment of other proteins, the insulin receptor substrates (IRSs), which recognize these phosphorylated tyrosine residues, known as phosphotyrosine binding (PTB) domains. After the recruitment of IRSs to the receptor, they are phosphorylated by the receptor on different residues. Then, phosphorylated IRSs acts as scaffolds proteins for other proteins, by Src-homology 2 (SH2) domains [16,17]. There are multiple IRSs proteins, which are involved in different functions depending on the tissue [18]. Furthermore, IRSs can be phosphorylated in serine and threonine as well, leading to attenuation in insulin signaling. Alternatively, there are other adaptor molecules, which could be recruited to the activated IR, including the Src homology 2 domain-containing (Shc) proteins. One of the critical steps after insulin treatment is the activation of phosphoinositide-3 kinase (PI3K) and Akt signaling pathway, which are recruited by the SH2 domains to the IRSs. After that, the catalytic domain phosphorylates phosphatidylinositol 4,5-bisphospate (PIP2), generating phosphatidylinositol (3,4,5)-triphosphate (PIP3), which is maintained in the cell membrane. Afterward, PIP3 recruits several proteins with pleckstryn homology (PH) domains, such as Akt and phosphoinositide-dependent kinase 1 (PDK1). Then, these proteins are recruited to the cell membrane, and Akt is phosphorylated in threonine 308 by PDK1 activity. It is known that insulin can phosphorylate Akt in Ser 473, which is mediated by another kinase, known as the mechanistic target of rapamycin complex 2 (mTORC2) [19]. Then, Akt is fully active and phosphorylates its target substrates, including FOXO, tuberous sclerosis complex 2 (TSC2), glycogen synthase kinase 3 β (GSK3β), Bad among others [18].

Alzheimer’s disease (AD) is a neurodegenerative disorder and one of the major causes of dementia in the world. Furthermore, AD is linked to T2DM being, at least, part of the mechanisms shared between these two diseases. One of the common features connecting both diseases is insulin resistance. Hence, the link between T2DM and AD is nowadays more and more evident, and molecular pathways that characterize this crosstalk are emerging because of the numerous pathophysiological homologies among both diseases. But there are still a lot of questions to be answered about how T2DM might influence AD and the derived therapeutic strategies that could be used to impulse future approaches to more efficient treatments.

## 2. Insulin Resistance

Insulin resistance is an altered response of insulin receptors to a given insulin concentration. This insulin resistance can occur at multiple levels of insulin signaling and is involved in different mechanisms. In this section, we are going to explain the main regulators of insulin resistance in cells in more detail.

### 2.1. MAPK and Insulin Resistance

The family of mitogen-activated protein kinases (MAPK) comprises 14 different components involved in the control of different cellular processes, such as proliferation, survival, differentiation, and apoptosis. MAPKs are divided into four different sub-classes, including extracellular signal-regulated kinases 1 and 2 (ERK1/2), c-Jun N-terminal kinase (JNK1-3), p38 (α, β, γ, and δ), and ERK5. These kinases have involved in the appearance of insulin resistance [20]. In addition, there is another group of MAPKs, known as atypical kinases (ERK3/ERK4, NLK, and ERK7). However, this group of proteins is poorly understood and needs more investigation. A recent review about all these groups of proteins was completed by the authors of [20].

#### 2.1.1. Extracellular Signal-Regulated Kinases 1 and 2 (ERK1/2)

After insulin receptor activation, IRSs are recruited and phosphorylated in tyrosine residues by the insulin receptor, leading to the recruitment of the adaptor protein growth factor receptor-bound protein 2 (Grb2). Alternatively, another adaptor protein Src homology 2 domain-containing (Shc) and then tyrosine phosphorylated in different residues [21]. Then, Shc binds Grb2, forming a complex with the guanine nucleotide exchange factor (GEF) called son of sevenless (SOS). SOS is a GEF for p21^Ras^, facilitating its activation. After that, Ras-GTP is active and phosphorylates the rapidly accelerated fibrosarcoma (Raf) [22], translocating to the cell membrane. Raf is a serine/threonine kinase, and is known as the MAP kinase kinase kinase (MAPKKK), which then phosphorylates and activates a MAP kinase kinase (MAPKK), the serine, and threonine MAPK/ERK kinases (MEK). Afterward, MEK phosphorylates the MAPK, extracellular-signal regulated kinase (ERK), regulating the phosphorylation status of different proteins in the cells [23,24,25]. The main effectors of ERK1/2 are by direct phosphorylation or by indirect mechanisms, including MAPK-interacting kinase (MNK), mitogen and stress-activated kinase (MSK), and p90 ribosomal S6 kinase (p90^RSK^). The direct effects are involved in the control of metabolism, such as gluconeogenesis, in the control of protein synthesis, as well as in the mTORC1 signaling pathway. Furthermore, ERKs can directly regulate lipid homeostasis through the control of the sterol regulatory element-binding protein 1α and 2 (SREBP1α/2) or the hypoxia-inducing factor (HIF1α), among others, and reviewed by the authors of [20,23]. Regarding the indirect effects, ERKs can regulate the phosphorylation state of the protein synthesis initiation regulator, eIF4E, through MNK activation [26,27]. MSK can phosphorylate the cAMP response element-binding protein (CREB) [28], and p90^RSK^ can regulate glycogen synthesis through the modulation of GSK3 phosphorylation [29,30].

One of the mechanisms involved in insulin resistance is through the modulation of insulin receptor clathrin-mediated endocytosis [31,32,33]. Very recently, it has been proposed that both Src homology phosphatase 2 (SHP2) and MAPK are implicated in the endocytosis of insulin receptors and reviewed by the authors of [34]. ERK proteins are one of the best-characterized groups of proteins able to phosphorylate IRSs [35,36]. IRS1 and IRS2 can bind directly to one of the clathrin adaptors called AP2M1, promoting insulin receptor endocytosis, and mediated by the concomitant action of both SHP2 and ERK proteins [37]. Furthermore, ERK proteins control insulin resistance in in vitro studies by the effect of c-Jun activation domain-binding protein-1 (JAB1), activated under chronic inflammation, increased in insulin-resistant states, and mimicked using palmitic acid in hepatocytes [38]. Furthermore, oxidative stress is related to insulin resistance in cardiac tissue through the downregulation of nuclear factor erythroid 2-related factor 2 (Nrf2) mediated by ERK phosphorylation [39]. In this regard, nicotine, which generates insulin resistance in the heart, inhibits Nrf2 expression with a concomitant increase in ERK phosphorylation status in this tissue [40]. ERK signaling can upregulate bromodomain-containing protein 2 (Brd2), a nuclear serine/threonine kinase, during adipocyte differentiation, generating insulin resistance [41]. In addition, Brd2 is involved in the negative control of adipogenesis through its action towards c/EBPα and PPAR-γ via ERKs [42]. Another possible mechanism for enhancing insulin resistance is through the increased phosphorylation status of β-3 adrenergic receptor in an ERK2-dependent manner, facilitating lipolysis [43]. Very interestingly, the elimination of kinase suppressor of Ras 2 (KSR2) in mice, a scaffold protein that coordinates Raf/MEK/ERK signaling pathway, causes insulin resistance and obesity [44]. Direct activation of PPAR-γ by ERK has been observed with a concomitant increase in insulin resistance, which could be suppressed by cyclin-dependent kinase 5 (CDK5) in a MEK-dependent manner [45]. However, tumor necrosis factor-α (TNF-α), involved in insulin resistance, stimulates CDK5 phosphorylation in an ERK-dependent manner [46]. A maintained ERK activation suppresses the expression of the gluconeogenic enzyme glucose-6-phosphatase (G6Pase), decreasing glucose output in liver cells, through the ERK-dependent phosphorylation and retention of FOXO in the cytosol [47,48]. Sustained activation of ERKs downregulates the expression of both insulin receptors and IRSs (IRS1 and IRS2), with a concomitant reduction in insulin signaling in adipocytes [49]. Tyrosine phosphorylation mediates the docking effect for downstream effectors and adaptor proteins, such as PI3K and SHP2, favoring insulin signaling. However, serine/threonine phosphorylation, depending on where it occurs, could have a different effect on insulin signaling [50]. In macrophages, it has also been described an enhanced production of cytokines in response to insulin by the modulation of ERKs, as well as the modulation of inhibitor of κB kinase β (IKKβ). The consequence of this dual activation is the serine-phosphorylation of IRSs [51]. Nuclear factor-kB (NF-kB), which is linked to inflammation and insulin resistance, modulates several cytokines, including interleukin-6 (IL-6) [52]. In this regard, it has been defined a protective role of p53 in improving insulin signaling via the inhibition of NF-kB and ERKs, although the exact molecular mechanism of p53 in insulin signaling is not understood [53]. Blocking ERKs activity promotes the development of obesity and insulin resistance depending on the animal model that it is used [54,55].

#### 2.1.2. c-Jun N-Terminal Kinase (JNK)

It is very well known that JNKs are activated in response to insulin [56,57,58]. Different studies indicate the involvement of JNKs in obesity-induced insulin resistance and T2DM [59,60,61]. The activation of JNKs after insulin stimulation is dependent on MKK4 and MKK7 [62]. Originally, JNKs were originally described as the activating kinase in the N-terminal domain of the transcription factor c-Jun [63]. As it was previously explained in the case of ERKs, JNKs can phosphorylate directly different targets, or indirectly by activating intermediary kinases, such as p90^RSK^ [57,64,65]. As the direct nuclear target is GR, involved in gluconeogenesis and perilipin in the control of lipolysis. In addition, indirectly, JNKs through p90^RSK^ activation can regulate glycogen synthesis as ERKs do [29,30].

JNKs are involved in the regulation of obesity, T2DM, and insulin resistance [66,67,68,69]. Although there are three proteins belonging to JNKs, only JNK1 and JNK2 have a negative effect on insulin signaling [70]. However, it seems that both kinases do not have overlapping functions. In this regard, mice with JNK1 deletion were protected from obesity and insulin resistance [71]. However, the deletion of JNKs protects from atherosclerosis [72]. In any case, it is known as crosstalk between JNK1 and JNK2 in the control of obesity and insulin resistance [73]. One of the most obvious mechanisms of insulin resistance generation is through the modulation of IRS1 phosphorylation status, suggested using in vitro approaches [74]. The involvement of JNKs in the prevention of insulin resistance depends on the tissue. For instance, JNK1 deletion in adipose tissue was protected against liver steatosis. Paradoxically, JNK1 ablation in hepatocytes developed insulin resistance [75]. In this regard, in knock-in mice with a mutation in which the authors replaced Ser 307 with Ala 307 in IRS1 resulted in increased insulin resistance [76]. Alternatively, ER stress is linked to obesity, insulin resistance, and T2DM [77,78]. In fact, the use of different chemical chaperones can reduce ER stress, diminishing insulin resistance [79]. In this regard, different compounds with a potentiating action on ER chaperone capacity, uncovering azoramide as a compound for protecting cells against ER stress [80,81]. A high fat diet (HFD) induces ER stress as well, leading to JNK1 activation [82]. Furthermore, saturated fatty acids activate the JNK signaling pathway, as well as a protein kinase C (PKC)-dependent mechanism [83]. The JNK-interacting protein 1 (JIP1), a scaffold protein that is able to interact with different components of JNK signaling, has a key role in the activation of JNK in the adipose tissue of obese mice [84]. Then, JIP1 exerts an essential role in the control of metabolic stress regulation of JNK activity, and the elimination of JIP1-mediated JNK activation leads to a decreased in obesity-induced insulin resistance [85] and associated as a potential regulator of T2DM in humans [86].

#### 2.1.3. p38

These groups of proteins are activated by different growth factors, inflammatory cytokines, hypoxia, oxidative stress, among others [25]. Although the mechanism responsible for p38 activation in response to insulin is unknown, insulin treatment has been associated with p38 in adipogenesis. p38 can either directly phosphorylate its targets or can mediate its effects indirectly, by the activation of MNK and MSK. Its direct effectors are, many of them, shared with the other MAPKs (JNKs and ERKs). For instance, FOXO, PPAR-γ, and C/EBP-α are regulated by both ERK and p38 MAPKs [87]. However, PGC1-α is regulated by p38 [88,89]. Furthermore, it is involved in the phosphorylation of many other substrates, including ATF2 [90] and C/EBP-β [91] involved in adipocyte differentiation. Indirectly, through MNK activation, p38 regulates the protein synthesis initiation factor eIF4E, and through MSK, controls CREB protein [92].

The activation of p38 in the liver from obese mice diminishes ER stress, through the phosphorylation of X-box binding protein 1 (XBP1) and its translocation to the nucleus [93]. Furthermore, through ATF6 phosphorylation, another component of the unfolded protein response (UPR), promotes its translocation to the nucleus and regulates transcription of the luminal ER-resident chaperone called glucose-regulated protein-78 (Grp78)/Bip [94]. p38 MAPKs are implicated in inflammation through the modulation of ATF2 and NF-kB [95,96]. Under oxidative stress, there is an increased in IRS-1 serine phosphorylation and associated with a decrease in IRS1 protein levels, by an increased degradation in rat muscles under oxidative stress conditions [97]. The use of a p38 inhibitor, improved insulin-dependent glucose transport, indicating a decrease in insulin resistance [98]. Moreover, p38 MAPKs, and more specifically, p38α regulates other receptors, such as epidermal growth factor EGF/ErbB family, belonging to the tyrosine-kinase receptors, facilitating IRS-1 phosphorylation and insulin resistance in response to different stimuli [99]. Furthermore, another two isoforms, p38γ and p38δ, have been associated with hepatic steatosis, controlling neutrophil infiltration [100]. Very interestingly, p38 activation in the hypothalamus, could regulate insulin resistance and inflammatory cytokines expression [101,102].

#### 2.1.4. ERK5

First of all, it is also known as big mitogen-activated protein kinase-1 (BMK1). ERK5 is controlled as all MAPKs upstream by different kinases. As MAPKKKs, it can be considered MEKK2 and MEKK3, phosphorylating specifically MEK5 (MAPKK), and then, ERK5. Like the rest of MAPKs, there are a great variety of substrates, including transcription factors (MEF2, Sap1, and others) and other kinases, such as the serum and glucocorticoid-induced protein kinase (SGK) [103,104]. Although it is necessary more knowledge, ERK5 activation mediates protection against diabetes and obesity, through different actions [20]. For instance, ERK5 induces an anti-inflammatory effect through the activation of PPARδ [105]. Furthermore, it is known that ERK5 is involved in insulin sensitivity in adipocytes [106]. Very recently, it has been published the implication of ERK5 in the preservation of IRS1 through the repression of *mir128-3p* under hypoxic conditions [107]. However, more efforts are needed to understand more deeply the effects of ERK5 in insulin resistance in the different tissues.

### 2.2. PI3K/Akt and Insulin Resistance

Phosphoinositide 3-kinases (PI3Ks) are a family of lipid kinases that phosphorylate phosphatidylinositol (PI) and its derivatives phosphatidylinositol 4,5-bisphosphate (PIP2) and phosphatidylinositol 4-monophosphate (PI4P) [108]. PI3Ks are divided into three classes: Classes Ⅰ, Ⅱ, and Ⅲ. The best-characterized is the class Ⅰ PI3K, which is a heterodimer composed of a catalytic subunit p110 and a regulatory subunit p85 or p84/p101, depending on the subdivision in classes ⅠA or ⅠB, respectively [109].

It is known that growth factors, cytokines, and hormones, activate receptor tyrosine kinases (RTKs), and then, their adaptor proteins GPCR (G-protein-coupled receptors) activate class I PI3Ks, which are previously recruited to the plasma membranes relieving the inactivation function of p85 and p110; indeed, RTKs and GPCRs can also activate Ras to activate PI3K [110]. Class Ⅰ PI3K phosphorylates PIP2 to form PIP3 on intracellular membranes. Phosphatases, such as phosphatase and tensin homolog (PTEN), act as negative regulators of PI3K, generating PIP2 from PIP3 through dephosphorylation. PI3K is activated to recruit downstream effectors, including AKT [111].

AKT/PKB is a serine-threonine kinase divided into three isoforms: AKT1 expresses ubiquitously contributing to the maintenance of glucose homeostasis and regulating adipogenesis [112]; AKT2 is mainly expressed in insulin-sensitive tissues, and AKT3 is located in the brain, regulating neuronal size and functions [113,114]. AKT is activated by initial phosphorylation in the kinase domain by phosphoinositide-dependent protein kinase 1 (PDK1) and subsequent phosphorylation in the regulatory domain through mTORC2 by a PI3K-dependent mechanism [115]. Since AKT is activated, several apoptotic and cell cycle substrates and signaling pathways related to glucose and lipid metabolism are regulated [111,116].

Both insulin and IGFs can bind to their respective receptors, leading to activation and autophosphorylation [117]. There are six isoforms of IRS proteins—IRS-1 and IRS-2 are the most studied and present a major contribution to PI3K signaling. Once IRS proteins are phosphorylated, catalytic subunit are activated, obtaining PIP3 for AKT [118].

AKT pathway is a critical regulator of metabolism. First of all, the activation of PI3K/AKT stimulates glucose mobilization through the activation of AS160/TBC1D4 [119], which promotes the translocation of glucose transporter 4 (GLUT4) and the inhibition of TXNIP, which controls the internalization of glucose transporters [118]. AKT can also regulate glycolysis by stimulating hexokinase and phosphofructokinase [117,120] coordinating gluconeogenesis and fatty acid oxidation through FoxO1 phosphorylation [121]. This phosphorylation results in its nuclear exclusion and inhibition of its activity in several tissues, by reducing gluconeogenesis and glucose levels through the reduction in PGC-1α [122]. When insulin signaling is activated, AKT also phosphorylates GSK-3β and inhibits its activity, leading to an increased glycogen synthesis [123]. AKT also phosphorylates TSC2 [117,118].

In the development of T2DM, insulin resistance occurs after chronic excessive energy conditions with a concomitant lipid accumulation and increased lipolysis with a hypertrophied adipose tissue [119,124,125]. Obesity-induced inflammation and the increased secretion of adipocytokines reduces IRS-1 activation [126], and the AKT-dependent glucose uptake and mediated inhibition of lipolysis [112,124,127].

The excess of free fatty acids (FFA) reduces blood adiponectin levels and lipid oxidation in other tissues, triggering lipotoxicity and insulin resistance [128]. Those FFA generated reduces glucose transport and glycogen synthesis in skeletal muscle, inhibiting IRS-1 activity [129]. This lipotoxic environment in skeletal muscle leads to an excessive production of reactive oxygen species (ROS) that activate intracellular stress kinases to also inhibit IRS-1 [130] and another downstream effector of the AKT pathway [131]. Circulating lipids can also contribute to peripheral insulin resistance and the impairment of PI3K/AKT in the brain [132]. Dysregulated secretion of leptin and the increased expression of FoxO1 in propiomelanocortin (POMC) neurons provoke hyperphagia and obesity [133,134], as well as hypothalamic hyperactivation of mTORC1, seems to induce hepatic insulin resistance through the inhibition of IRS-1/AKT axis and K (ATP) channels [135,136].

In the pancreas, PI3K-mediated insulin secretion is blocked with a deleterious effect on pancreatic β cells [137,138]. In the liver, excessive oxidation of circulating FFA hyperactivates pyruvate carboxylase activity and gluconeogenesis via FoxO1 [139], while DAG increased levels reduce PI3K/AKT signaling to exacerbate insulin resistance [140]. Thus, glycogen synthesis is inhibited, and insulin-mediated de novo lipogenesis (DNL) stimulated [141,142].

### 2.3. mTOR and Insulin Resistance

The mechanistic or mammalian target of rapamycin (mTOR) is an evolutionary conserved serine/threonine protein kinase which is related to the PI3K kinase family (PIKK) [143] and exists in two different complexes: mTORC1 and mTORC2 [144,145], known as “rapamycin-sensitive” and “rapamycin-insensitive”, respectively [144].

The structure of both complexes shared common proteins that include mTOR itself, mammalian lethal with sec-13 protein 8 (mLST8 or GβL), DEP domain-containing mTOR-interacting protein (DEPTOR), Tel two-interacting protein 1 (Tti1), and telomere maintenance 2 (Tel2) [146,147,148]. Specifically, mTORC1 contains Raptor (regulatory-associated protein of TOR) and PRAS40 (proline-rich Akt substrate 40kDa) [149,150]; whereas, mTORC2 core components are Rictor (rapamycin-insensitive companion of mTOR), mSIN1 (stress-activated protein kinase-interacting protein 1), and protein observed with Rictor 1 and 2 (PROTOR 1/2) [151,152,153]. Figure 1 depicts the composition of both mTOR complexes in cells.

Focusing on mTORC1 activation, when growth factors bind to its receptors, PI3K is activated and phosphorylates Akt, phosphorylating TSC2. TSC2 interacts with TSC1 [154], acting as a GTPase activating protein (GAP) for Rheb [155]. Rheb is a small G protein located on the surface of the lysosome, and its GTP-bound form directly stimulates the kinase activity of mTORC1 [144,145]. TSC1-TSC2 complex converts the active form of Rheb into its inactive GDP-bound state, negatively regulating mTORC1 [156]. Akt-mediated TSC2 phosphorylation inhibits its GAP activity causing mTORC1 activation [156]. Moreover, Akt phosphorylates PRAS40, which acts as a mTORC1 inhibitor, causing its dissociation from Raptor and mTORC1 activation [157].

Alternatively, AMPK, phosphorylates TSC2, boosting TSC1-TSC2 association and causing mTORC1 inhibition [158]. Moreover, there is also a TSC2-independently mTORC1 modulation through AMPK by Raptor phosphorylation that leads to mTORC1 inactivation [159]. In addition, glycogen synthase kinase 3β (GSK3β) phosphorylates TSC2 acting as a negative regulator of mTORC1, but the Wnt pathway activates mTORC1 by inhibiting GSK3β [160]. Another mTORC1 modulation occurs under low-glucose conditions by Rheb sequestration, provoking mTORC1 inhibition [161]. Lastly, amino acid activates mTORC1 through Rags, a group of small G-proteins similar to Rheb [162].

The best-characterized substrates of mTORC1 are eukaryotic translation initiation factor 4E (eIF4E) binding protein 1 (4E-BP1) and S6 kinase 1 (S6K1), which promote protein synthesis and ribosome biogenesis [163]. In addition, mTORC1 positively controls the synthesis of lipids [164]. Moreover, adipogenesis is regulated through the higher activity of peroxisome proliferator-activated receptor γ (PPARγ) caused by mTORC1 [165]. Other studies reported that mTORC1 positively regulates mitochondrial biogenesis and oxidative metabolism by modulating PPAR-γ coactivator 1α (PGC1-α) [166], and cellular metabolism and ATP production by activating HIF1α transcription [167].

One of the most important pathways that are regulated by mTORC1 is autophagy, and the biogenesis of lysosomes. For instance, mTORC1 directly phosphorylates unc-51-like kinase 1 (ULK1), which is required to initiate this process, provoking its inhibition [168].

Although mTORC2 was originally described as rapamycin-insensitive, in the last few years, it has been observed that the long-term treatment with rapamycin suppresses mTORC2 activation [169]. While mTORC2 is insensitive to nutrients, this complex responds to growth factors, such as insulin through, possibly, ribosomes. This potential mechanism suggests that mTORC2 binds to ribosomes and explains that these organelles may be needed for mTORC2 activation [170].

What is well described is the relationship between mTORC2 and some members of the AGC subfamily of kinases, including Akt, SGK1, and protein kinases C-α (PKC-α). Akt is fully activated through phosphorylation by the kinase activity of mTORC2 [169]. Lastly, mTORC2 positively regulates PKC-α activation, which is involved in cell shape through changes in the actin cytoskeleton [151].

Although both complexes of mTOR are involved in the pathogenesis of diabetes through their actions in β-cell metabolism and immune cells [171], the role of mTORC1 in insulin resistance and in the progression of T2D will be explained in more detail for a better understanding of type 3 diabetes (T3D).

Active mTORC1 causes a decrease in glycemia, hyperinsulinemia and improves glucose tolerance in mice, which can be reverted by rapamycin [172]. mTORC1/S6K1 pathway was identified as a regulator of β-cell apoptosis, size, and autophagy, whereas mTORC1/4E-BP2-eIF4E pathway seemed to control β-cell proliferation [173]. Although mTORC1 positively controls β-cell function, a sustained activation in mice leads to higher insulin resistance in the pancreas, reducing cell survival and promoting apoptosis. In addition, IRS1 phosphorylation and inactivation cause downregulation of insulin sensitivity [174]. Firstly, mTORC1 increases β-cell mass but, upon aging, exists a loss of β-cells causing hyperglycemia and hypoinsulinemia [175].

Furthermore, mTORC1 promotes adipogenesis through the activation of S6K1, which controls the expression of early adipogenic transcription factors, causing adipose tissue expansion, the main risk factor for developing insulin resistance, and thus, of T2DM. Moreover, 4E-BPs are activated by mTORC1 and regulates the final differentiation of adipocytes through PPAR-γ [176].

This complex is highly active in the adipose tissue and in the skeletal muscle of obese mice, causing insulin resistance through inhibition of insulin signaling by the S6K1 pathway, reducing glucose uptake by the muscle, and contributing to systemic insulin resistance [177]. In the liver, this high activation promotes hepatic insulin resistance through the degradation of IRS1, contributing to the dysregulation of glucose and lipid homeostasis [178,179]. Moreover, a hyperactivation of mTORC1 in the heart muscle causes a well-known complication of T2D [180].

At the prediabetic or early stages, mTORC1 positively regulates pancreatic β-cells’ growth and secretion of insulin, but when mTORC1 remains chronically overactive, the compensatory insulin secretion mechanism does not work effectively, pulling to β-cell death. 

## 3. Autophagy

Autophagy is a catabolic cellular mechanism that participates in the maintenance of cellular homeostasis by degrading different cellular components. Three different types of autophagy have been described: Microautophagy, chaperone-dependent autophagy (CMA), and macroautophagy, the latter being the most common.

### 3.1. Types of Autophagy

In microautophagy, invaginations are formed in the lysosomal membrane, where the components to be removed are located [181]. In CMA, the hsc70 chaperone recognizes the specific aminoacidic sequence KFERQ in cytosolic target proteins, binding to them along with other cochaperones. These aggregates reach the lysosome membrane, where they are recognized by LAMP-2A monomers. LAMP-2A monomers multimerize, forming complexes that facilitate the protein’s entry into the lysosome where it is degraded [182]. Macroautophagy (hereafter as autophagy), is a highly conservative cellular process and consists of several well-characterized stages: The formation of the phagophore, initiation of the encapsulation of the element to degrade (nucleation) carried out mainly by VPS34 and Beclin-1 complex [183], and fusion with the lysosome, which results in the degradation of the engulfed components [184].

Autophagy can be classified according to the cargo to degrade, being the most studied and characterized autophagy of mitochondria or mitophagy. This system is one of the main quality control mechanisms of mitochondria by degrading damaged mitochondria [185]. By the canonical pathway, when a mitochondrion is depolarized, it facilitates the recruitment of PINK1 to the outer mitochondrial membrane, being accumulated [186]. The accumulation of PINK1 induces the recruitment of Parkin, which will ubiquitinate different proteins, such as mitofusins [187]. This ubiquitination causes the elimination of these proteins, through proteasome, as well as the formation of the autophagosome around the damaged mitochondria, through p62. This protein can directly interact with the lipidated form of the LC3B, which is the link between the autophagosome and damaged mitochondria [188].

### 3.2. Autophagy in Pancreatic β Cells: A Double-Edged Sword

In the pancreatic β cell, although in the short term, the inhibition of autophagy may exert a beneficial effect by stimulating insulin secretion, in the long term, an endoplasmic reticulum (ER) stress could be generated, causing cell death [189]. In this regard, inhibition of autophagy makes the pancreatic β cells more susceptible to ER stress damage [190]. In murine models, the loss of autophagy generates an impairment in glucose-induced insulin secretion; a decreased in β cell mass and hyperglycemia in pancreatic β cells [191], as well as an increase in both ER stress and insulin resistance, in the liver [192]. On the other hand, autophagy is the main mechanism by which human islet amyloid polypeptide (hIAPP) is degraded, a polypeptide whose accumulation in β cells produces cellular toxicity [193,194]. Recently, the use of autophagy inducers has been shown to improve the metabolic profile of hIAPP-overexpressing mice, by reducing the accumulation of oligomers of IAPP [195].

Interestingly, the overexpression of hIAPP in pancreatic β cells disrupts lysosomal membrane integrity and altered lysosome-dependent degradation [196]. In accordance with this, the overexpression of hIAPP diminishes both the viability and survival of pancreatic β cells both in vitro [197] and in patients with prediabetes or T2DM [198]. The latter is important, since the accumulation of damaged mitochondria prevents the correct intracellular Ca^2+^ traffic, altering insulin secretion in β cells or insulin signaling in target tissues [199]. This is because the networks of interactions between the membranes of the mitochondria and those of the ER, called mitochondria-associated ER membranes (MAMs), are altered [200], where different proteins of the insulin signaling pathway have been located as mTORC1 [201] and PTEN [202].

In recent years, different investigations have been analyzed the possible beneficial effects that autophagy activation has on the development of T2DM in murine models. Rapamycin administration, an inhibitor of the mTORC1 pathway, causes functional failure in the islets [203]. While in insulin target tissues (muscle, liver, and adipose tissues), the response to insulin improves, in β cells, an impairment in glucose tolerance has been observed, by chronic activation of autophagy, inducing an increase in insulin granules degradation [189,204]. On the other hand, it has been seen that the administration of an inducer of autophagy through the AMPK-ULK1 pathway, improves both glucose tolerance and insulin sensitivity in high fat diet (HFD)-fed mice [205]. 

## 4. Endoplasmic Reticulum (ER) Stress

The ER is a membrane network localized in the endoplasm [206]. Nuclear ER presents an outer, and an inner membrane with a lumen between them joined at the nuclear pores involved in protein synthesis, maturation, and trafficking [207]. Once there, proteins are folded by chaperones [208] and suffer from posttranslational modifications, such as glycosylation or disulfide bond formation [209,210]. After maturation, only correctly folded proteins are exported to Golgi apparatus, being the unfolded or misfolded proteins processed by ER-associated degradation (ERAD) machinery [211], and finally degraded by the proteasome [212]. Other important functions of the ER are the storage of calcium, regulated by the sarco/endoplasmic reticulum calcium ATPase (SERCA) pump, among other components [213], and the contribution to lipids biosynthesis like ceramides, phospholipids, or cholesterol [214].

### 4.1. The ER Stress Response

ER stress is a condition where misfolded proteins cannot reach their native folding state, due to an increase in the workload or an inefficient degradation through the ubiquitin-proteasome system, and then aggregate in the ER lumen [215]. The mammalian ER stress response has four mechanisms regulated by different pathways:

#### 4.1.1. PERK Pathway

The protein kinase RNA (PKR)-like endoplasmic reticulum kinase (PERK) is a type Ⅰ transmembrane protein that senses the accumulation of unfolded proteins in the ER lumen [216]. When the UPR is activated, BiP protein releases from PERK, and PERK response depends upon the changes in chaperone level, specifically GRP-78 [217]. Both proteins associate in a complex that leads to PERK autophosphorylation, and then PERK inactivates the eukaryotic initiation factor 2 α (EIF2α), essential for protein synthesis [218]. Attenuation of EIF2α also triggers the translation of ATF4, a transcription factor involved in amino acid metabolism or the resistance to oxidative stress, essential for the stress response [219]. 

#### 4.1.2. IRE1 Pathway

IRE1 is a transmembrane protein with a dual activity. During ER stress, IRE1 is activated and promotes the splicing and expression of X-box binding protein 1 (XBP1) mRNA, increasing ERAD components, ER heat shock proteins, and lipid biosynthesis. Later, due to its RNAse activity, IRE1 introduction into the ER lumen is reduced [220].

#### 4.1.3. ATF6 Pathway

ATF6 is a type Ⅱ transmembrane protein responsible for the sensing of unfolded proteins. In response to the accumulation of unfolded proteins and the ER stress, BiP dissociates from ATF6, and then it is transported to the Golgi apparatus, where it is cleaved and the cytoplasmic portion obtained translocates into the nucleus. Thus, ATF6 activates the transcription of ER chaperone genes [214,221].

#### 4.1.4. Apoptosis-Inducing Pathways

If the previously mentioned pathways cannot suppress the toxic effect of ER stress, cell death pathways are triggered; the most known are described below. C/EBP homologous protein (CHOP) is a transcription factor induced by the activation of ATF4 and ATF6, which leads to the expression of proapoptotic factors, such as ER oxidorreductin (ERO1) or growth arrest and DNA damage 34 (GADD34) [222]. Meanwhile, IRE1 forms a complex with tumor necrosis factor receptor-associated factor 2 (TRAF2) and apoptosis signal-regulating kinase 1 (ASK1), which phosphorylates JNK and triggers cell death. It is also known that caspases apoptotic cascade is also involved in ER stress [223].

### 4.2. ER Stress in Type 2 Diabetes Mellitus

During ER stress, the activation of the IRE1 pathway is essential for the synthesis and maturation of insulin, but hyperactivated IRE1 results in β-cell death through the initiation of JNK and caspases cascade [224,225]. The PERK-EIF2α-CHOP axis acts as a switch between a correct function of pancreatic β-cell and its survival ability. The inactivation of PERK has demonstrated a loss of insulin secretion and failure of the pancreas [218,226]. Meanwhile, ATF6 suppresses glucose formation via gluconeogenesis through CREB-regulated transcription coactivator 2 (CRTC2) in the liver [227].

Interestingly, it has been demonstrated that MAMs are essential in regulating ER stress and autophagy. Ca^2+^ flux into mitochondria through the MAM tethering complex increases ROS generation, which further promotes Ca^2+^ flux to the matrix by oxidizing the mitochondrial Ca^2+^ uniporter, resulting in an accumulation of redox species at the mitochondria-ER interface [215]. Consequently, this environment triggers UPR machinery and antioxidant mechanisms that contribute to the disruption of MAM in several tissues, increasing insulin resistance [228]. The main regulators in the control of insulin secretion by pancreatic β cells are depicted in Figure 2.

Moreover, the high amount of FFAs secreted, due to insulin resistance processes in several tissues, activate all the mechanisms of the UPR and advanced glycation end products (AGEs) also directly or indirectly induce an ER stress response [229].

## 5. Neurotransmitters

Neurotransmitters are a set of biomolecules of different nature synthesized inside neurons that allow communication from one neuron to another, from a neuron to a muscle cell, or from a neuron to a gland, also called neurotransmission or synaptic transmission. Once the neurotransmitter is synthesized inside the neuron, it is stored in vesicles that are released by exocytosis at the synaptic terminal. These neurotransmitters travel to the target cell, where they are recognized by membrane receptors, triggering two different types of responses: Ion channel opening (altering the polarity of the membrane) or the release of intracellular messengers. The signal ends when the neurotransmitter is degraded by enzymes or reabsorbed by the presynaptic neuron [230].

### 5.1. GABA Alterations in DM2

γ-Aminobutyric acid (GABA) is an aminoacidic neurotransmitter that acts as an inhibitor of the central nervous system in mammals [231]. High levels of GABA have been found both in diabetic rats [232] and in patients with T2DM [233], in which they observed less cognitive performance. The authors hypothesized that this effect of GABA could be due to its role as a negative inhibitory function on dopamine release in the mesocortical dopamine pathway, which partly projects to prefrontal cortex. In this regard, it has also been observed that in patients with T2DM, there is a higher concentration of GABA in the medial prefrontal cortex, which causes a decreased memory for face-occupation associations [234], which highlights the negative effect that high concentrations of GABA have on the brain of patients with T2DM. Despite this, the cause for which there is more GABA in T2DM patients remains unclear, although it is thought that in part, it may be because in these patients, there are alterations within the GABA–glutamate–glutamine cycle [233]. However, more studies should be carried out to clarify this question.

### 5.2. Effects of Dopamine on Insulin Secretion

Dopamine is a neurotransmitter that is included in catecholamines, and unlike GABA, it can send both activating and inhibitory signals depending on the type of cell receptor they activate. Dopamine has been shown to have a regulatory effect on glucose-induced insulin secretion in β cells through dopaminergic D2 receptors, whose deletion in mice causes a decrease in pancreatic β cell mass, decreased β cell replication, impairment of insulin response to glucose, high fasting glucose levels, and glucose intolerance [235]. Interestingly, at low concentrations, dopamine has an activating effect on insulin secretion, while at high concentrations has inhibitory effects, which highlights the regulatory role of this neurotransmitter in insulin secretion capacity [236]. Moreover, in the diabetic situation, an increased turnover of dopamine to norepinephrine in the pancreatic islets have been previously reported [237], being very important, since high concentrations of norepinephrine have an inhibitory effect on the secretion of insulin in the β cells, preventing the uptake of dopamine [238].

## 6. Human Amylin Misfolding and T2DM

As previously mentioned, the main player in T2DM development is the survival of pancreatic β-cell, which mainly relies on the capacity to counteract the huge demand for insulin secretion to maintain glucose homeostasis. But in this compensatory effect, there is an ambiguous effect, due to pancreatic β-cells also increase production of amylin, a hormone cosecreted with insulin. Amylin (or islet amyloid polypeptide, hIAPP) is a 37 aa peptide mainly implicated in food intake regulation and short-term satiation, exerting its action upon pancreatic β-cells and several brain areas [239]; recently, it has been described to cooperate with leptin action, favoring neurogenesis [240] and participating on hedonic control of eating [241]. It is secreted inside insulin granules, and its maturation and folding process occurs within endoplasmic reticulum (ER), being this a critical step. First insights unraveled hIAPP as the main component of islet amyloid extracellular deposits in response to hyperglycemia [242], but there are described many mechanisms involved in hIAPP-mediated intracellular toxicity in pancreatic β-cells, particularly those affecting the cellular protein turnover.

### 6.1. hIAPP and ER Stress in Pancreatic β-Cells

The continuous hyperglycemia forces pancreatic β-cells to increase insulin production until excessive rates for ER capacity. Both transient and chronic hyperglycemia (>16.7 mM glucose plasmatic levels) trigger in pancreatic β-cells ER stress, leading to failure in the protein maturation process and consequently misfolded both insulin and hIAPP accumulation in the ER lumen [218,243]. More precisely, misfolded pro-hIAPP and hIAPP accumulation may trigger a sustained ER stress, and therefore, the associated apoptosis, when this overload leads to an excessive activation of PERK, the consequent perinuclear expression and nuclear translocation of CHOP plus its inhibitory activity upon anti-apoptotic Bcl-2, as well as an increased alternative-splicing of XBP1 [244,245,246]. hIAPP toxic effects on pancreatic β-cells have been inhibited by chemical chaperones or other molecules via ER stress alleviation [247,248,249]. It has also been described that not only hIAPP, but also merely high glucose levels or palmitate might induce ER stress and mediate pancreatic β-cell apoptosis, mainly by increasing ROS levels in ER, and therefore, compromising the correct folding process, which feeds a dangerous vicious cycle [250]. Under these conditions of accumulating misfolded hIAPP, increasing oxidative stress in ER environment, and even hypothetic ER membrane damage by hIAPP toxic oligomers, there could be a Ca^2+^ influx from ER to mitochondria, endangering mitochondrial integrity, due to indirect ROS-mediated oxidative stress [251].

#### Autophagy as Defense Against hIAPP Aggregates

Autophagy, as previously described, is a vital process in cell survival, even though more important when harmful elements (such as toxic protein oligomers) are present in the cytoplasm. Therefore, autophagy is essential to maintain pancreatic β-cell homeostasis [190,252]. This process takes even more relevance regarding the toxicity of hIAPP aggregates. Higher rates of LC3B-Ⅱ and p62 accumulation in the cytoplasm of pancreatic β-cells expressing hIAPP have been described specific p62 silencing by shRNA has shown to increase caspase-3 cleavage exacerbating hIAPP toxicity [196]. In addition, Atg7 deficient mice presented increased apoptosis and less β-survival [194]. Recent studies demonstrate that overexpression of hIAPP in INS1E cells leads to an mTORC1 hyperactivation and impaired autophagy [197]. Furthermore, there is an additional hypothesis about the ability of misfolded proteins, such as α-synuclein or hIAPP, to damage lysosomal membranes, disrupting the autophagy flux and triggering apoptosis [253]. Thus, evidence point to the role of hIAPP oligomers in disrupting their own clearance by autophagy, enhancing their toxic potential. It has been observed that autophagy-inducing compounds, such as rapamycin and resveratrol, could modulate hIAPP aggregates level by autophagy activation [193,254]. A new detoxifying system has been described by our laboratory; it has been proved that hIAPP aggregates could be included in MVB and be secreted by exosomes in order to counteract pancreatic β-cells cytoplasmic hIAPP aggregates accumulation and cell death [255]. 

### 6.2. hIAPP Damage on Mitochondria

Special regard should be made in terms of mitochondria. As it has been previously described, mitochondrial homeostasis is vital in pancreatic β-cells, due to their energetic requirements and the consequent mitochondrial abundance in this cell population. For maintaining a healthy mitochondrial pool, continuous mitochondrial turnover should function properly, so mitophagy and its molecular mechanisms are critical in the pancreatic β-cell. The toxic properties of hIAPP, for example, in terms of membrane disruption by its interaction with negative charges of membrane lipids, would mediate mitochondrial dysfunction and apoptosis induction of these cells [256]. In addition, loss of mitochondrial membrane potential (ΔΨm), ATP production and mitochondrial mass, and subsequent caspases activation is detected in INS1-E cells exposed to hIAPP [257,258]. Mitochondrial functionality also depends on a good balance between mitochondrial dynamics, fusion, and fission, as has been explained before. The fission state of mitochondria leads them to mitophagy, through specific protein machinery that labels damaged mitochondria to be degraded by autophagy. It has been demonstrated that INS1E cells overexpressing hIAPP show in a basal situation fissioned mitochondria and a defective elimination by mitophagy, accumulating dysfunctional mitochondria in the cytoplasm [197]. 

### 6.3. The Inflammatory Response to hIAPP Aggregates in T2DM

The immune system also has a role in the hIAPP-mediated pancreatic β-cell failure. It has been described as a complex signaling network regulating the immune response to hIAPP aggregates accumulation in pancreatic β-cell. The cytokine IL-1β is described as the main mediator of inflammation upon pancreatic β-cells, and several studies have described how the protein machinery needed for IL-1β maturation and secretion, called inflammasome, is activated in T2DM [259]. The inflammasome is composed of a nucleating pattern recognition receptor (PRR), which is mostly the protein of the Nod-like receptor family (NLR), NLRP3. In addition, an adaptor protein (the apoptosis-associated speck-like protein containing a CARD, ASC) and caspase-1 activity are needed to complete the inflammasome activation.

There are two necessary steps to finally activate IL-1β and induce inflammation; the first one is known as priming, consisting in the production of enough mRNA levels of NLRP3 and pro-IL-1β, and is triggered by the “signal 1”, composed by a heterogeneous group of events known as danger-associated molecular patterns (DAMPs). In the case of T2DM, this signal might be minimal-oxidized LDL (mmLDL), free fatty acids (FFAs), reactive oxygen species (ROS), lysosomal dysfunction, or even altered cytoplasmatic ionic flux by membrane disruption. It has been described as “signal 2”, which is necessary to fully activate the inflammasome, misfolded proteins known as causative factors of most important neurodegenerative diseases [260].

A lot of studies have been developed to unravel the role of hIAPP and islet amyloid deposits in inflammasome activation and inflammation on T2DM. It has been demonstrated that hIAPP aggregation triggers NLRP3-mediated inflammasome activation, caspase-1-mediated IL-1β release, and macrophage and dendritic cells recruitment to islets both in vivo, in hIAPP-transgenic mice islets [259,261,262,263,264,265] or human islets [266], and in vitro models [267]. In addition, most of these publications indicate that not hIAPP fibrils, but hIAPP oligomers are responsible for inflammasome activation, and even it has been evidenced that hIAPP could activate NLRP3 in a ROS-independent manner, although this molecular mechanism needs more research [268]. Interestingly, recent studies highlighted hIAPP, and its receptors could mediate microglia inflammasome activation in Alzheimer’s disease (AD) mice [269,270], and these receptors could even mediate neuroinflammation mediated by amyloid β peptide (Aβ) in vitro [271].

Therefore, T2DM is a disease whose pathophysiological features affect many molecular aspects of metabolic pathways and inflammatory response. In association with obesity, its non-stop growing incidence has encouraged many studies about not only new therapeutic strategies, but also to emphasize the potential influence of T2DM on other disease development. A special mention is required to be made about neurodegeneration; more than 34 million people around the world are affected by two of the most prevalent neurodegenerative disorders, AD [272] and Parkinson’s disease (PD) [273]. From several years ago, there are many studies pointing to epidemiological evidence of a higher risk of AD in prediabetic or type 2 diabetic patients [274,275,276,277]. Thus, molecular and pathophysiological links between both diseases are being researched to establish possible diagnostic and therapeutic common strategies. 

## 7. Molecular Pathophysiology of Alzheimer’s Disease

AD is considered the major cause of dementia worldwide, although it is more prevalent among western and developed countries [278]. It is typically classified into familial AD (fAD) or sporadic AD (sAD), depending on the pathological etiology. fAD is associated with the presence of some autosomal-dominant mutation in key genes, such as amyloid β precursor protein (AβPP), the γ-secretase catalytic components presenilin-1 (PSEN1) and presenilin-2 (PSEN2), or apolipoprotein E (apoE-ε4); this AD type only represents the ~2% of cases [279]. In sAD, almost 98% of cases have a more complex etiology, combining genetic predisposition, epigenetic events, and environmental factors. Most sAD patients are elderly people with many other comorbidities that may cooperate with sAD events and affect the course of the disease [280]. Among these comorbidities, obesity, cardiovascular pathologies, such as stroke and T2DM, are thought to be the most prone to cooperate or aggravate sAD development. Understanding the molecular pathways implicated in AD and T2DM and how they could be related may establish more efficient diagnostic protocols for type 2 diabetic patients and is unraveling new therapeutic targets to avoid neuronal and pancreatic β-cell death. 

### AD Pathologic Characteristics: The Role of Amyloid β and tau

AD is characterized by cooperative key players in its course: Aβ neuronal plaques, tau neurofibrillary tangles, and neuroinflammation [281]. The most known and prominent feature of AD is the Aβ plaques derived from the alternative processing of amyloid precursor protein (APP). APP is a transmembrane protein ubiquitously found in the central nervous system (CNS); there are different species of Aβ according to the amino acid (aa) composition, being the most abundant in the brain are the 40 and 42 aa (Aβ_40_ and Aβ_42_) [282]. 

The APP protein is processed by two different and mutually excluding pathways: The secretory pathway and the amyloidogenic pathway. The secretory pathway is begun by C-secretase-mediated APP cleaving, being released a soluble N-terminal fragment (sAPPα) and a C-terminal fragment (C83); the latter is then cleaved by γ-secretase to origin a smaller C-terminal fragment (C3).

These secretase activities have been attributed to the ADAM (a disintegrin and metalloprotease family) proteins—more precisely, α-secretase activity seems to be related to ADAM-7 and ADAM-10 [283]. The amyloidogenic pathway starts with the β-secretase (also known as a β-site amyloid precursor protein (APP)-cleaving enzyme 1, BACE1)-mediated cleavage of APP, releasing a smaller N-terminal fragment (sAPPβ) and a longer C-terminal containing the full amyloidogenic aminoacidic sequence (C99). Next, cleavage by γ-secretase will origin the amyloid β (Aβ) peptides.

These peptides are extremely amyloidogenic, and these monomers form rapidly oligomers, prefibrillar aggregates, and finally, fibrils that will form β-amyloid plaques; it has been demonstrated that Aβ_42_ is the most prone to aggregation, so it is the most neurotoxic peptide [284]. Initially, extracellular Aβ amyloid plaques and Aβ fibrils were thought to play the main role in the amyloid pathogenic pathway of AD, but recent insights have highlighted Aβ oligomers as the most toxic species [285]. Aβ_42_ oligomers trigger many deleterious effects on the neuronal environment, which cooperate to enhance neurotoxicity and neuronal cell death.

Neurofibrillary tangles, the other hallmark of AD, are composed of hyperphosphorylated tau protein aggregates. Tau is a microtubule-assembly protein required to the correct assembly of microtubule proteins and intracellular trafficking, being critical in neurons for axonal activity maintenance. The toxic potential of tau protein is due to its phosphorylation status: However, normal tau is a phosphoprotein and required to be phosphorylated to exert its function.

In AD patients’ brains, tau is found to be hyperphosphorylated ~3-fold more than normal brain’s tau. A complex balance between tau kinases and phosphatases is required to maintain this delicate homeostasis, and some of these enzymes are being tested as potential therapeutic targets. Phosphorylation sites of tau have been widely searched in this process (more than 40 have been identified), being probably phosphorylated by the combined action of proline-directed protein kinases (PDPKs) and non-PDPKs. GSK3β and CDK5 have been pointed as the main kinases acting upon AD-tau phosphorylation sites, but there is evidence of cooperation among several kinases to regulate tau phosphorylation. Both in vitro and in vivo, prephosphorylation of tau by PKA promotes additional phosphorylation by GSK3β; therefore, sustained activation of PKA could be relevant at the initial stages of AD.

Regarding the other side of the balance, phosphatases that act upon tau might play an important role in counteracting its hyperphosphorylation. Phosphoprotein phosphatase 2A (PP2A) is one of the few phosphatases targeting tau protein with a wide tissue distribution and high specificity in tau regulation. It has even been described that activation of GSK3β inhibits PP2A activity and inhibition of PP2A activates GSK3β, highlighting the complex environment of kinases and phosphatases involved in tau regulation. Hyperphosphorylation of tau drives to the formation of paired helical fragments (PHFs), which will form the neurofibrillary tangles. In addition, AD-tau inhibits normal-tau binding to tubulin, impairing its activity upon microtubules. PHFs have been shown to impede the trafficking of neurotrophins, and therefore, impair axonal and dendritic transport. In addition, overexpression of tau has been demonstrated to affect morphology, cell growth, and especially, transport of organelles mediated by microtubule-dependent motor proteins. Surprisingly, there are studies showing that hyperphosphorylation of tau may protect neurons from apoptosis, but the loss of function triggered by AD-tau creates a propitious situation for other AD insults to drive to neuronal death [286]. 

## 8. The Relation Between T2DM and AD: A Molecular Approach

### 8.1. Brain Insulin Resistance and Its Impact on AD

Related to T2DM, brain insulin resistance also develops a crucial role in AD pathogenesis. Insulin and insulin-like growth factor Ⅰ and Ⅱ (IGF-Ⅰ, IGF-Ⅱ) have an important role in the brain, as these peptides and their receptor genes are widely expressed in neurons and glia, including neurodegeneration-targeted structures [287]. Insulin is known to regulate gene expression and trafficking of glucose-transporter 4 (GLUT4), critical for brain glucose metabolism, and decreased levels of GLUT4 expression have been found in postmortem AD patients’ brains [288]. Both insulin and IGF have been reported to mediate neuron and glial growth, metabolism, survival, gene expression and protein synthesis, neurotransmitter network, and synaptic functionality [289].

Clinical studies have shown a link between brain lower insulin receptor levels and IGF-I signaling deficiency [288,290,291]. In addition, more precise data have unraveled molecular mechanisms whereby insulin might influence Aβ aggregates managing; its action may favor insulin-degrading enzyme (IDE) and ADAM-10, and decrease BACE1, GSK3β, and APP gene expression, promoting Aβ extracellular release [292]. It has also been proved that insulin deficiency triggers Aβ_42_ and amyloid plaques accumulation, increased tau phosphorylation and neurofibrillary tangles formation and spatial memory impairment [293], and even Aβ inhibits insulin-mediated JNK/TNF-α signaling by promoting IRS-1 phosphorylation and impairing insulin signaling [294], thus feeding a vicious cycle. Furthermore, defects on IDE, due to mutations in its gene, drive to decreased enzymatic clearance of insulin and Aβ, and consequent hyperinsulinemia and Aβ accumulation [295]. Interestingly, proteotoxic accumulation of Aβ aggregates have been shown to interfere with neuronal insulin signaling cascade by competition with insulin, and even modify sensibility and surface expression of IR [296,297].

Insulin resistance has also been related to the tau pathology of AD. There are many preclinical studies relating not only T2DM, but also type 1 diabetes with increased tau phosphorylation [298,299,300], and increased levels of tau have been found in CSF of type 2 diabetic patients compared with normal patients [301]. The role of insulin and IGF-Ⅰ action upon tau mediated by GSK3β activity has shown to be preferential, since several years ago [302]. In addition, impaired insulin signaling results in PI3K-Akt and Wnt-β-catenin pathway decreased activity, and activation of GSK3β, thus promoting tau phosphorylation [298,303]. The defective insulin signaling also affects other kinases that may contribute to tau hyperphosphorylation, such as p38 and JNK, probably together with an impaired PP2A activity [304]; ERK 1/2, another kinase of the insulin signaling pathway, has also been related to tau phosphorylation under oxidative stress conditions [305]. In conclusion, there are hallmarks relative to T2DM systemic insulin resistance and brain insulin resistance characterizing AD’s onset and progression, which point to possible shared pathologic mechanisms. Both molecular mechanisms of insulin resistance and their consequences are summarized in Figure 3.

### 8.2. Hyperglycemia and Its Consequences on AD Pathologic Development

In addition to insulin and IGF-Ⅰ signaling impairment, the hyperglycemic status of type 2 diabetic patients might mediate specific effects on AD pathogenesis. Hyperglycemia affects neuronal homeostasis: It has been reported that repeated transient hyperglycemic episodes, such as it happens in prediabetic or type 2 diabetic patients, could affect K_ATP_ channels, altering neuronal metabolism and increasing Aβ levels [306]. In addition, aging and dysregulated glucose metabolism drives the uncontrolled and non-enzymatic reactions between sugars and lipids, free amino groups of proteins and nucleic acids, giving, as a result, advanced glycation end-product (AGEs).

AGEs are more abundant in AD patients with diabetes than in non-diabetic AD patients, and they have been proved to impair Aβ_42_ normal clearance and favor Aβ and tau glycation, which promote amyloid plaques and neurofibrillary tangles formation [307]. Specifically, glyceraldehyde-AGEs (glycer-AGEs), which have been shown to be the most extended AGEs, are increased in diabetic patient’s serum and have enhanced toxicity on neurons [308]. There is evidence showing that AGEs specific receptors (RAGEs), which are found in neurons, microglia, astrocytes, and vascular endothelial cells, interacts not only with AGEs, but also with Aβ and mediate inflammatory effects [309]; it is even described that RAGEs could mediate Aβ transport through blood brain barrier (BBB) or enhance expression of BACE1, thus promoting Aβ formation [310]. It has been proved that AGEs could induce tau hyperphosphorylation through RAGEs-GSK3β signaling activation [311], and direct AGEs injection in mice brain displayed AD pathological features, such as decreased memory and increased tau phosphorylation, APP expression, and Aβ_42_ formation [312,313]. 

### 8.3. The Impact of ER Stress in AD 

As in T2DM, ER homeostasis is also a critical point in neurodegenerative disorders, such as AD. The most known role of ER stress in neurodegenerative disorders is related to proteostasis of protein aggregates featuring AS, PD, or HD, but there is growing evidence for the implication of ER stress in other neurological pathologies [314]. As it has been mentioned, ER response to stress is through activating the coordinated pathways of the unfolded protein response (UPR).

In contrast to T2DM, ER stress is not a direct cause of the pathological cascade of events characterizing AD, but an important contributor to its development and aggravation [315]. Apart from its activation upon damaging events of AD (Aβ, p-tau), some UPR effector proteins have been shown to develop an important role in synaptic plasticity and maintenance of cognitive function. Contrary to what happens in T2DM, Aβ_42_ and p-tau seem to be the major contributors to ER stress. Neuronal cells exposed to Aβ_42_ have shown a dramatic ER Ca^2+^ deregulation and a consequent ER protein misfolding [316], maybe due to the interaction of Aβ neuronal N-methyl-D-aspartate receptors (NMDARs) [317]; even in some fAD cases, there is evidence for Aβ_42_ accumulation within ER, triggering the activation of UPR [318]. Aβ could even bypass ER-mediated UPR rescue by inhibiting proteasome, thus inducing chronic ER stress-mediated apoptosis in cooperation with JNK activation [319]. Generally, Aβ peptide direct administration or genetic expression drives to increased levels of eIF2α phosphorylation, Bip, CHOP, GADD34, and cleaved-caspase 12 genetic expression [320]. PERK, one of the UPR early effectors, could induce BACE1 accumulation through its protein synthesis inhibitory effect; mRNA of BACE1 contains some upstream open reading frames that trigger its higher translation upon eIF2α phosphorylation [321].

More precise experiments have been conducted, and they have shown that Aβ oligomers injected directly in hippocampus trigger-specific PERK activation, induce ATF4 expression in axon and CHOP-mediated neuronal apoptosis [322]. To confirm its role in AD progression, lack of PERK and diminished eIF2 phosphorylation reduced Aβ levels and impedes defects in memory [323]. The downstream effector of UPR, XBP1s, also has a complex role in AD. Its function affects relevant genes in APP processing, such as PS1 and PS2, APP trafficking mediators, or ADAM-10 [324,325]; there is also evidence that the role of XBP1s in posttranslational modifications of BACE1 [326].

In addition, decreased levels of the E3 ubiquitin ligase HRD1 are found in AD, being this key enzyme expression regulated by XBP1s [327]; HRD1 not only has a role in managing ubiquitinated proteins for its degradation (such as Aβ), but also could target BACE1 or intervene in APP expression [326,327]. Altogether, this evidence highlights the importance of the UPR mediator XBP1s in the Aβ pathway of AD. In terms of tau phosphorylation and neurofibrillary tangles formation, ER stress also seems to have a contributing role. There has been found a correlation between Bip overexpression and increased activity of GSK3β and high levels of phosphorylated tau, pointing to possible regulation of GSK3β activity through Bip under ER stress conditions [328]. PERK activation and consequent eIF2α phosphorylation in the hippocampus region has been found to colocalize with aberrant tau phosphorylation [329]. The main player of the other branch of UPR, IRE1α, has also been related to p-tau and GSK3β stimulation; phosphorylated IRE1α is found in AD’s patients’ brain, and diminished of IRE1α activity displayed abolition of GSK3β activation, so IRE1α seems to be an essential pathway for its activity [330,331]. XBP1s is also a target key gene in tau homeostasis, such as tau kinase Cdk5 or the above-mentioned HRD1, which can tag p-tau to be degraded by the proteasome [324,332]. 

### 8.4. The Relevant Role of Mitochondria on AD Progression

As it happens in T2DM, mitochondria are a basic pillar in neuronal homeostasis and a priority target in AD. The high energy expenditure displayed by neurons, in addition to their weakness in facing oxidative stress, point to mitochondria as one of the most vulnerable organelles in neurons: The oxidative phosphorylation (OxPhos), carried by electron transport chain (ETC) protein complexes, is the major source of ROS.

Mitochondrial dysfunction and the consequent alteration in its bioenergetic role are some of the earlier and most important characteristics in AD progression, even before Aβ or p-tau. Disruption of glycolytic processes, impairment of ETC enzymatic reactions, increased ROS and defective antioxidant mechanisms are a common hallmark between AD patients; in fact, defects in glucose metabolism are used as a predictive marker to foresee the progression of the disease [333,334]. Apart from its role in metabolism and ROS production, Ca^2+^ homeostasis exerted by mitochondria together with ER becomes critical in terms of neurons, and dysregulation of this intracellular ionic balance has severe consequences not only in cell homeostasis, but also in synaptic mechanisms of neurotransmission [335,336].

It seems that mitochondrial dysfunction occurs before Aβ aggregates, but then this aggregation will cooperate in mitochondrial failure. Mitochondrial translocase of the outer membrane (TOMM) and the previously mentioned RAGEs have been shown to mediate incorporation and aggregation of Aβ in mitochondria; RAGE knockout mice neurons displayed a protective effect from Aβ by diminished uptake [337,338]. In addition, Aβ could also disrupt the function of ETC, through its binding to the heme groups found in complex Ⅳ that cooperate as redox centers in OxPhos, thus compromising ATP production [339]. Even APP has been demonstrated to affect mitochondrial protein import machinery, complexing both translocase inner mitochondrial membrane 23 (TIMM23) and TOMM40 and interrupting cytochrome c function [340,341]. The mitochondrial permeability transition allows ion diffusion from matrix to cytoplasm, and plays an important role in cytochrome c release and apoptotic factors; a key component of the mitochondrial permeability transition pore (mPTP), Cyclophilin D (CypD), increase its translocation to the inner membrane by Aβ action, triggering mPTP opening, increasing ROS production and affecting mitochondrial calcium buffering capacity [342,343].

These events could feed a vicious cycle, due to the evidence pointing to aberrant processing of APP and hyperphosphorylation of tau promoted by mitochondrial defective activity [344,345]. Hyperphosphorylated forms of tau have also been described as harmful elements for mitochondrial homeostasis. P-tau has shown to affect directly ETC complex Ⅰ, therefore triggering increased ROS production, lipid peroxidation, and decreasing the antioxidant activity of enzymes, such as superoxide dismutase (SOD) [346]. An association between p-tau and VDAC mitochondrial pore protein is also reported, impairing its function by blocking it [347]. This evidence points to a bidirectional disrupting relation between tau and mitochondria in AD. However, not only is mitochondrial functionality compromised in AD, but also mitochondrial dynamics and their recycling through mitophagy is impaired.

Insights from several studies revealed that excessive mitochondrial fission status over fusion in AD’s patients’ brains; it has been demonstrated that Aβ could interact with the fission protein DRP1, promoting the increased free radical production that hyperactivates DRP1 and Fis1-mediated mitochondria fragmentation, and consequently, failed mitochondrial location to synapses, decreased ATP synaptic production and overall synaptic dysfunction [348]. In the same direction, an interaction between p-tau and DRP1 has been described, enhancing excessive fission [349].

To complete the failure of mitochondrial homeostasis, there is also evidence of defective mitochondrial biogenesis: Both in AD’s patients’ brain, AD cellular models, and APP and tau mice models of AD, reduced levels of mRNA encoding critical players in mitochondrial biogenesis (such as PGC1α, Nrf-1, Nrf-2, and TFAM) have been found [350,351,352]. Increased mitochondrial fission is also thought to be promoted by AGEs, maybe through increased expression of DRP1 and Fis1, as well as downregulate fusion proteins expressions, such as Mfn1, Mfn2, and Opa1. Regarding mitophagy management in AD, all the above-mentioned events may drive long-term fissioned mitochondria to elimination, so it is clear the importance of the correct development of mitochondrial recycling; however, dysfunctional mitophagy is another common feature found in AD progression. It has been observed that Parkin-mediated mitophagy is more induced by Aβ aggregates and aberrant tau than the non-canonical mitophagy pathway [353]. There is evidence that Parkin is absent in AD neurons cytosol, thus promoting an abnormal accumulation of PINK1 in the outer mitochondrial membrane (OMM) and the accumulation of defective mitochondria [354].

Some studies hypothesize that Parkin is aberrantly recruited to mitochondria in association with the ubiquitin-C-terminal hydrolase L1 (UCHL-1) and truncated tau protein, leading to an excessive mitophagy that compromises synaptic functionality [355]; other insights point to the fact that Parkin is sequestered by pathologic tau in the cytosol, impeding its recruitment to OMM and compromising correct mitochondrial elimination [356]. Lysosome’s disruption observed in AD could also provoke defective mitochondria accumulation in axons, damaging synaptic processes by interrupting Ca^2+^ correct influx [357].

### 8.5. mTOR Hyperactivation and AD

mTOR develops a critical function in neuronal metabolism and is also a widely studied target in AD. The main function regulated by mTOR to integrate signals for cellular growth and proliferation, so it becomes even more relevant in terms of neurons. mTOR plays its main role in neurogenesis through enhancing expression of brain-derived neurotrophic factor (BDNF), and it has shown to be a key regulator of neuronal functionality, as the principal mediator of axonal and dendritic growing and regeneration not only during development, but also in the mature nervous system [358]; through different signaling encoding, mTOR is capable of counteracting axonal damage both in the central nervous system (CNS) as in peripheral nervous system (PNS) [359]. Memory consolidation, memory recall, and synaptic plasticity have demonstrated to be necessarily mediated by mTOR [360,361]. As is seen in T2DM, mTOR hyperactivation is a pathological situation that may aggravate or facilitate neurological pathologies [362,363] and neurodegenerative disorders, such as AD [364,365].

As previously discussed, insulin resistance is a common hallmark of T2DM and AD, and it is mainly driven by the PI3K/Akt/mTOR signaling pathway [366]; more precisely, neuronal insulin resistance driven mTOR hyperactivity inhibits IRS1 activity by negative feedback. It is a consensus that brain insulin resistance in AD is promoted by a chronic mTOR signaling pathway hyperactivity, pointing to this metabolism master regulator as a critical target for therapeutic research. There are many studies about mTOR implication in AD that test its role by the most known inhibitor, rapamycin. Rapamycin has been shown to mitigate mTOR hyperactivity, even in a diet-induced insulin resistance scenario [365,367,368]; furthermore, the induction of insulin resistance and mTOR hyperactivation exacerbate cognitive decline and AD pathogenesis [369].

Interestingly, this event is shared by T2DM, and it may be one of the theoretic links between both pathologies [298,370]. It has been demonstrated that insulin negative regulates Aβ deposition and tau phosphorylation, so mTOR hyperactivation and consequent insulin signaling disruption increase the impact of these events [371]. In addition, Aβ has been shown to decrease not only the expression of neuronal insulin receptors [372], but also IRS1 phosphorylation on Ser^307^ [373], thus breaking the linkage between IRS1 and IR. IRS1 phosphorylation status has also been reported to be altered by p-tau [374]. mTOR has also been shown to partially mediate neuroinflammation in AD. All the impaired metabolism via mTOR-mediated insulin and IGF-Ⅰ resistance trigger not only oxidative stress, but also inflammatory response in neuronal environment. Several models have demonstrated that mTOR could regulate neuroinflammation, and mTOR chronic hyperactivation is linked to systemic inflammation [375,376]; however, most of the studies relating mTOR to inflammation have been conducted in acute neuroinflammation models. It has been proved that mTOR inhibition could decrease the expression of proinflammatory cytokines TNF-α, IL-1β, and IL-6, and reduced caspase-3 activation, after brain hemorrhage [377]. mTOR action upon inflammation is not only limited to neurons: mTOR inhibition drives to reduced microglial response to cytokines and influence macrophage by preventing M1-inflammatory polarization and promoting anti-inflammatory M2 type [378,379]. Neuroinflammation has been linked with neuronal insulin resistance, due to the TNFα-JNK pathway-mediated IRS1 inhibition. Both hallmarks of AD, Aβ and misfolded tau, have been shown to activate TNF-α [380].

Altogether, these results show the central role of mTOR in the crosstalk between brain insulin resistance, neuroinflammation, and AD progression. In addition, mTOR also develops a crucial function in synaptic plasticity. Synaptic plasticity is the ability of synapses to strengthen or weaken in response to an increase or decrease in their activity. Enhancing the number of neurotransmitters released or their postsynaptic receptor expression are the main actions aimed at adapting the cells’ response to neurotransmission; the two different processes playing that role are long-term potentiation (LTP) and long-term depression (LTD).

Both processes need high rates of protein expression, which are thought to be mediated by mTOR downstream effectors 4EBP and p70S6K [360,381]. NMDA receptor and BDNF upregulate mTOR signaling in a critical way for LTP control and stimulation [382,383]. mTOR has also shown important for metabotropic glutamate receptor (mGluR) in the context of synaptic plasticity: mGluR-mediated LTD is associated with increased phosphorylation of p70S6K and S6 [384]. TSC1/TSC2 complex, an mTOR key regulator, also seems to be essential in the hippocampus for synaptic plasticity [385]. In addition, there has been found Aβ injection impaired synaptic plasticity in association with mTOR signaling, confirming the mediating role of this protein complex in Aβ-mediated synaptic dysfunction [381]. Another neurotransmitter homeostasis is also altered in AD pathogenesis: Aβ has shown to conduct dysfunction and loss of GABAergic inhibitory interneurons [386,387], although more studies are needed to unravel the molecular mechanisms underlying this alteration. 

### 8.6. Relevance of Autophagy in AD Neuronal Homeostasis 

Autophagy, which is under mTOR regulation, has been considered as one of the most important cellular processes in AD, as well as a promising target in therapeutic strategies. Autophagy impairment has been demonstrated to be a common hallmark of neurodegenerative disorders (AD, PD, HD, or ALS) characterized by cytoplasmic, extracellular, or nuclear inclusions and protein aggregates accumulation [388]. The impaired balance between autophagosome formation and autophagic flux clearance is typically found in AD, both for autophagosomes accumulation [389] or even excessive autophagic flux [390]. Reduction of beclin-1, an autophagy effector, in an AD mice model has resulted in Aβ accumulation, neuronal abnormalities, and apoptosis [391]; this supports the finding of reduced beclin-1 mRNA levels in AD patients’ cortex. Accumulation of Aβ and cognitive defects were reversed by the use of rapamycin to stimulate autophagy [392].

In addition, in AD has been observed an accumulation of autophagic intermediates and failed autophagosome maturation, thus impairing Aβ clearance [393]. The formation of amyloid plaques by extracellular Aβ is also influenced by autophagy [394]. Tau pathologic phosphorylation and accumulation could also be counteracted by autophagy. There are studies affirming that rapamycin could revert aberrant tau phosphorylation and prevent neurofibrillary tangles formation [365,395]; several insights strengthen the hypothesis about p-tau clearance by autophagy induction [396,397]. Deficiencies in the lysosome degrading role are also associated with Aβ and tau impaired clearance and pathological accumulation, since its fusion with autophagosome and proteolysis action is the final step of the autophagy degrading system [398]. Dysfunctional endosomal-lysosomal trafficking impedes this fusion and provokes autophagosomes accumulation upon AD’s patients’ brain [389,399].

Genetic studies have shown that presenilin-1 mutations, due to their role in lysosome acidification and activation of the lysosomal cathepsin protease, trigger defective lysosomal turnover of Aβ [400]. In summary, all of these results evidence that autophagy is a critical process for maintaining neuronal cytoplasmic homeostasis, supported by many studies using other autophagy enhancers for improving neuronal clearance in AD models [401,402]. Among these compounds, resveratrol has been widely used for AD damage treatment and may be a future therapeutic candidate [403,404,405]. Melatonin, a pineal hormone that regulates body response to circadian rhythms, has also recently been shown to activate autophagy, and therefore, to play a protective role in neurodegenerative disorders [406]. This autophagic impairment could result in aberrant proteins managing by other recycling systems, such as the MVB-exosome pathway. Exosomes, as it happens in T2M, are also a rising field of study in AD. Much evidence points to exosome-mediated clearance of Aβ aggregates and p-tau; however, as an opposite way as it happens in pancreatic β-cells, it seems that exosome detoxification of harmful proteins in AD contributes to the pathological spreading of the disease, as it is proved that exosome secretion inhibition leads to improving brain degeneration [407,408,409,410]. 

### 8.7. Inflammation as a Harmful Fuel in AD

As relevant as it is on T2DM pathogenesis, neuroinflammation might play one of the most determinant roles in AD progression and prognosis. In the case of neuroinflammation, the innate immunity mediators in the brain are astrocytes and microglia. A lot of evidence from postmortem brains of AD’s patients show many inflammatory cytokines, chemokines, and prostaglandins levels increased [411,412]. It seems that astrocytes and microglia are both able to express PRRs that allows them to initiate an inflammasome signaling cascade to ultimately activate IL-1β release through caspase-1 activation. In a healthy brain, low levels of IL-1β are found, probably due to the powerful inflammatory reaction that it can trigger; however, high levels of IL-1β are usually measured in AD patients’ brains [413].

Several years ago, it was first described NLRP3-mediated inflammasome activation, as well as IL-1β maturation and secretion, in AD by Aβ. Increased extracellular Aβ phagocyted by microglia may cause lysosomal disruption and cathepsin B release to the cytoplasm, triggering the inflammasome activation [414]. Further studies in AD mice models have confirmed the role of Aβ upon inflammasome signaling cascade initiation, as the NLRP3 genetic inhibition has demonstrated to protect from Aβ deposition and memory impairment, in addition, to establish Aβ duality as cause or consequence of inflammasome activation [415]. Furthermore, amyloid plaques have been reported to be able to secrete neurotoxic factors and recruit microglia, which in response secrete cytokines amplifying the inflammatory response throughout surrounding tissues [416]. All these effects have been shown to lead microglia to adopt a chronic M1 proinflammatory phenotype, which in turn promotes the Aβ deposition and memory impairment. In contrast, microglia-specific NLRP3 inhibition leads to an M2 anti-inflammatory phenotype, lower Aβ extracellular accumulation, and improved synaptic function [416,417]. In addition, elevated levels of IL-1β are able to induce tau hyperphosphorylation, affecting synaptic plasticity by LTP inhibition [418,419]. 

In summary, AD has shown to be a complex multifactorial disease, with critical events that might cooperate in the early onset of the disease and worsen its progression. It is not clear the molecular mechanisms that begin neuronal homeostasis disruption and trigger the pathological cascade characterized by amyloid β and aberrant tau deposition, and exacerbated neuroinflammation. The hypothesis of AD as a metabolic disorder is strongly growing, supported on neuronal glucose metabolism disruption impact and brain insulin resistance influence observed in the development of the disease [371]. 

## 9. The Crosstalk Between T2DM and AD: The “Type 3 Diabetes Mellitus”

The strong correlation between T2DM and AD, reflected as two to five times increased T2DM patient’s probability of developing AD [277,420], is also strengthening this theory, establishing common links between both pathologies and pointing to critical processes by which the diseases may influence each other [421]. As it has been explained, T2DM insulin resistance is a growing event from the prediabetic status that compromises pancreatic buffering activity upon hyperglycemia, and this insulin resistance is also extended to peripheral tissues, such as the brain.

The severe impact of brain insulin resistance not only on neuronal glucose metabolism (critical for their functionality), but also on the Aβ and tau pathological evolution highlights how “invasive” and relevant T2DM course could be upon cognitive impairment and AD onset. The uncontrolled hyperglycemia sustained during T2DM is also a participant in this cooperative development, since this abnormal status triggers AGEs extended formation, which helps to aggravate aberrant protein pathology and neuroinflammation of AD. Furthermore, the extended oxidative stress provoked by hyperglycemia has a especially damaging impact on the compromised neuronal homeostasis, thus deepening in the cell failure. If these injuries were not enough, the above-mentioned effects have catastrophic consequences on mitochondrial function, disrupting its metabolic function, exacerbating ROS production, and unbalancing Ca^2+^ delicate homeostasis, which in turn not only affects intracellular milieu, but also strongly compromises synaptic functionality. Effects of Aβ and tau upon mitochondrial dynamics and mitophagy complete this disastrous vicious cycle.

The common feature of mTOR hyperactivation in both diseases related to insulin resistance is another likely candidate for being a molecular link between these pathologies, causing in pancreatic β-cells and neurons not only dysregulation of insulin IRS-IR signaling pathway, but also intracellular quality control mediated by autophagy [145,370,422]. In addition to all these shared events among T2DM and AD, there is consensus about a pathologic link between both diseases: The well-known amyloidogenic potential of hIAPP upon pancreatic β-cells and its relevant influence on AD pathophysiology. Since many years ago, hIAPP has focused the attention of researchers, due to its emerging role on T2DM pathogenesis [423,424], and the similarities of its aggregation kinetics compared with neurodegenerative-causing proteins, such as Aβ or even α-synuclein, fed a wide group of hypotheses about the interaction among these peptides [423]. Recent evidence relates hIAPP with cognitive decline [425], and it has been found in the brain and cerebrospinal fluid of both T2DM and AD patients [426]. But the strongest evidence about hIAPP role in AD is the crossseeding with Aβ. It has been demonstrated that hIAPP is able to assemble with Aβ in vitro, being aromatic residues of the hIAPP the thought amyloidogenic core; both peptides do colocalize on brain amyloid deposits [426,427,428]. Derived from these results, hIAPP mimics are being investigated as Aβ anti-aggregative strategies [429]. In addition, there are studies affirming that peripheral-produced hIAPP could bind to its receptors on BBB’s endothelial cells, enhancing lipoprotein receptor-related protein-1 (LRP1) translocation and promote brain Aβ transport into the bloodstream [270]; this could trigger insulin signaling impairment by Aβ upon most insulin-sensitive tissues [430].

Even the cooligomerized hIAPP-Aβ complexes exhibit 3-fold more toxicity compared with single aggregates, not only in neurons, but also in pancreatic β-cells [430,431]. There is recent evidence that points to a bi-directional effect, since there has been observed cytoplasmic Aβ and tau inclusions within pancreatic β-cells in AD, but not T2DM patients [432]. Furthermore, ongoing hyperamylinemia in T2DM has been shown to mediate neurotoxic effects through its receptor, AMY_R_, by interacting with the non-selective cation channel TRPV4 and triggering uncontrolled Ca^2+^ influx [433,434]; it is thought that Aβ may also have the same effect, but this is still unraveled. The recent study published by our laboratory [256] opens a new hypothesis about crosstalk between pancreatic β-cells and AD development via exosome transport of hIAPP aggregates towards brain structures, such as hippocampus, although more research is needed to clarify this possibility upon T2DM mice models or human samples. In addition to the cooperative self-aggregation with Aβ, hIAPP may exacerbate other AD hallmarks aggravating its prognosis: hIAPP aggregates could increase ER stress (as it has been proved in pancreatic β-cells) already existing in neurons, compromising Ca^2+^ homeostasis and UPR, and inducing ER stress-mediated apoptosis. The hIAPP damage upon mitochondrial homeostasis would exacerbate neuron-local ROS production, contribute to Ca^2+^ imbalance and decrease oxidative phosphorylation and ATP production; even mitochondrial dynamics and mitophagy impairment could be caused upon already damaged AD-mitochondria. Likely, the possibility of inflammatory response exacerbation mediated by exogenous hIAPP aggregates in the neuronal environment is one of the most worrying links between T2DM and AD, because of the common inflammasome activation occurring in both disorders. The molecular hallmarks involved in the dysfunction of either pancreatic β cells in T2DM (Figure 4) or neurons in AD (Figure 5) are shown. 

## 10. Conclusions and Future Directions

In this review, we have analyzed the main signaling pathways, which connect, T2DM and AD at the molecular level. These mechanisms include insulin resistance, generated in part by the chronic low-grade inflammation during the progression of T2DM by different possible mechanisms (pro-inflammatory cytokines and hIAPP, among others). In addition, oxidative stress is produced by the generation of AGEs and by mitochondrial dysfunction. Then, mTOR contributes to insulin resistance, inducing ER stress as well. The accumulation of fissioned mitochondria is a consequence of altering mitochondrial dynamics with the concomitant activation of oxidative stress.

The accumulation of amylin aggregates disrupting the main mechanisms of cell proteostasis, such as ubiquitin-proteasome system and autophagy, and the potentiation of the activation of mTOR, generating a vicious cycle of insulin resistance and autophagy inhibition, in a scenario with an accumulation of altered and dysfunctional mitochondria. All of these mechanisms are contributing to pancreatic β cell failure and hence, to the appearance of multiple alterations in multiple tissues, including the brain. As a new mechanism, we have uncovered that when all these events occur in pancreatic β cells, alternative mechanisms are activated to eliminate amylin aggregates by non-canonical pathways, such as exosome production.

Our lab has demonstrated that, using an in vitro approach, pancreatic β cells can generate exosomes-bearing amylin aggregates, contributing to detoxify β cells from the aggregates but, at the same time, can favor amylin deposition in other regions of the body. This strategy opens a new avenue of a possible connection between the pancreas and the brain, contributing to a better understanding of the communication among different organs and tissues. However, many questions remain unanswered in this field. From our point of view, the most relevant ones are the following: (1) Is insulin produced by the brain? (2) Is the BBB damaged in the progression to T2DM? (3) What are the molecular mechanisms, or the factors involved in the induction of brain insulin resistance in T2DM? (4) Is enough insulin resistance to produce dementia or the appearance of AD? Hence, the link between T2DM and AD is nowadays more and more evident, and molecular pathways characterizing this crosstalk are emerging because of the numerous pathophysiological similarities and common pathogenic mechanisms between both diseases are being studied. But there are still many questions to be answered about how T2DM might influence AD and the derived therapeutic strategies that could be useful for better and more efficient therapeutic approaches.

## Figures and Tables

**Figure 1 cells-10-01236-f001:**
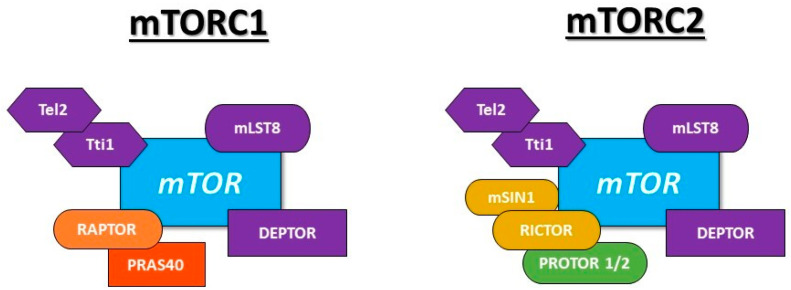
Structure of mTOR complexes. Both shared mTOR that is the catalytic subunit, DEPTOR, which acts as a negative regulator, Tti1/Tel2, as scaffold proteins, and mLST8, which function is unknown. Specifically, mTORC1 is also formed with RAPTOR (scaffold protein) and PRAS40 (inhibitor of RAPTOR); mTORC2 with RICTOR and mSIN1 both acting as a scaffold; PROTOR 1/2 is a positive regulator.

**Figure 2 cells-10-01236-f002:**
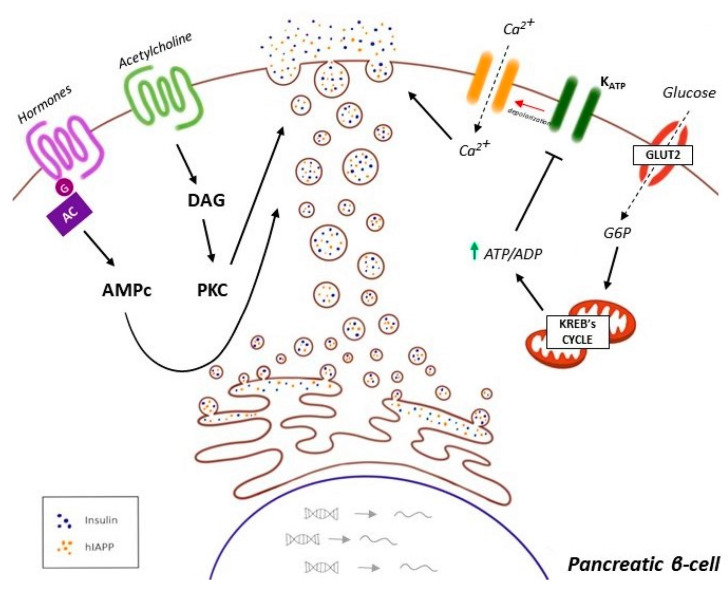
Mechanisms involved in insulin and hIAPP cosecretion in pancreatic β-cell. The most important physiological stimulus of insulin secretion is glucose. Glucose is transported inside the cell thanks to GLUT2, entering the Krebs Cycle and glycolysis. These processes produced a higher ATP/ADP rate that inhibits ATP-dependent K^+^ channels, depolarizing the membrane. Changes in the potential of membrane open Ca^2+^ channels, introducing Ca^2+^ into the cell and promoting the release of the insulin into blood flow after glucose intake. Not only this pathway releases this hormone, but also an increment of AMPc via apetite’s hormones signaling and the activation of PKC, due to the adrenergic response. What is more important is that amylin or hIAPP is cosecreted in these insulin granules after its maturation in ER.

**Figure 3 cells-10-01236-f003:**
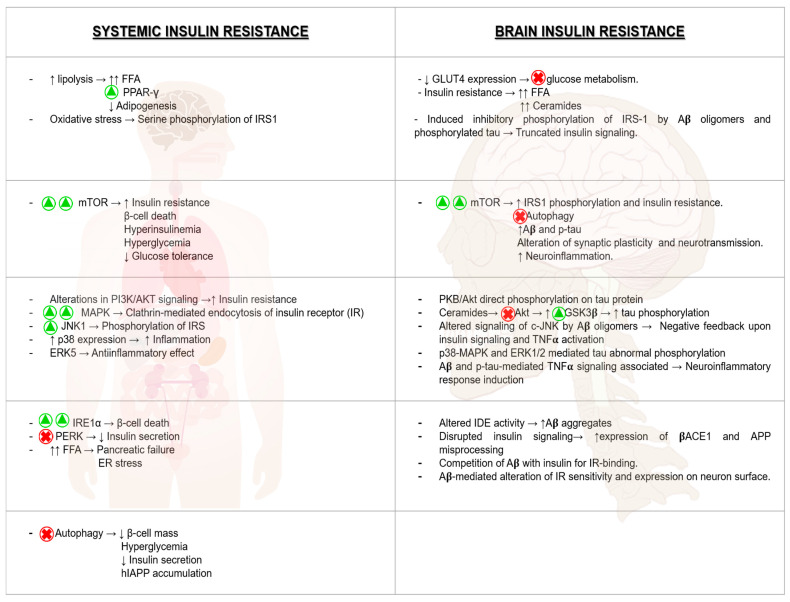
Molecular mechanisms of both systemic and brain insulin resistance and their consequences.

**Figure 4 cells-10-01236-f004:**
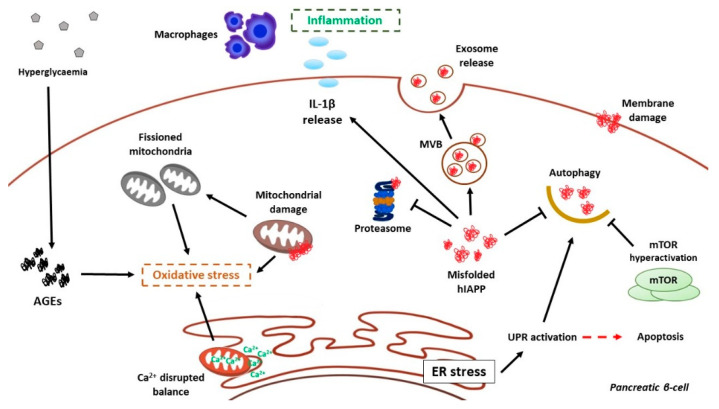
Hallmarks of pancreatic β cell failure in T2DM. Insulin secretory burden drives ER stress and misfolding of hIAPP. ER stress induces UPR activation in order to counteract protein aggregation, but chronic UPR activation leads pancreatic β-cell to apoptosis. ER stress could also promote Ca2+ unbalance among ER-mitochondria, increasing ROS and oxidative stress. Autophagy is also activated by UPR for hIAPP aggregates elimination, and its failure could promote hIAPP accumulation and detoxification via MVB-exosome secretion; pancreatic β-cell mTOR hyperactivation (due to insulin resistance or hIAPP) in turn also impaired autophagy flux. In addition, hyperglycemia and hIAPP direct interaction are thought to inhibit proteasome aggregates clearance. hIAPP mitochondrial damage could increase ROS production, oxidative stress, and accumulation of fissioned mitochondria. Hyperglycemia causes the formation of aberrant glycated molecules, the advanced glycation-end (AGEs), which increase oxidative stress. hIAPP aggregates could also induce NLRP3-induced inflammasome activation, IL-1β release, and macrophage recruitment.

**Figure 5 cells-10-01236-f005:**
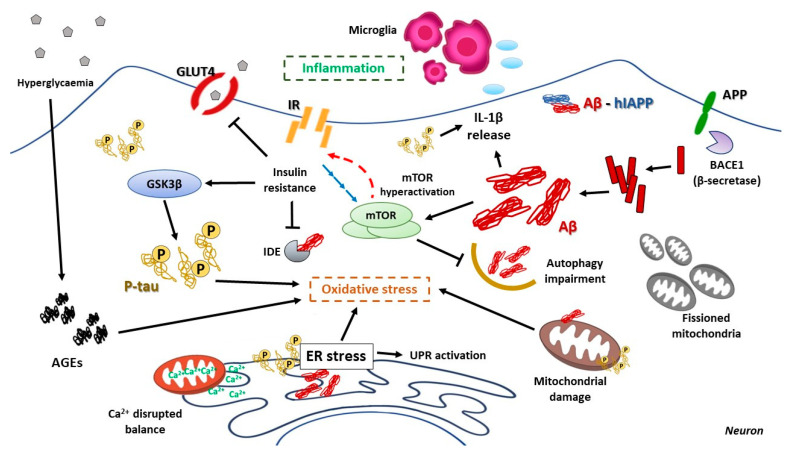
Hallmarks of neuron failure in AD. Neuronal alternative processing of APP through β-secretase (BACE1) origins intracellular Aβ40-42 aggregates that will origin extracellular amyloid plaques. In parallel, microtubule-associated protein tau protein aberrant phosphorylation drives to cytoplasmic neurofibrillary tangles formation. Brain insulin resistance reduces GLUT4 glucose transporter activity and decreases insulin-degrading enzyme activity (IDE); Aβ aggregates impaired insulin and IGF-I signaling cascade. Insulin resistance results in GSK3β activation and tau hyperphosphorylation. The hyperglycaemic status also induces AGEs which are known to impair Aβ clearance and promote GSK3β-mediated tau phosphorylation. Both proteins can induce ER stress, chronic UPR activation, and neuronal apoptosis. Aβ and p-tau are known to disrupt the mitochondrial respiratory chain, increase ROS, alter Ca2+ balance and lead mitochondria to an irreversible fission status; furthermore, Parkin-mediated mitophagy is impaired in AD by Aβ and p-tau. As it happens in T2DM, insulin resistance-mediated mTOR hyperactivation affects insulin signaling, impairs neurogenesis and synaptic plasticity, and impede correct autophagy-mediated Aβ and p-tau degradation; both peptides could also affect the lysosomal function and compromise autophagosomes clearance. Aβ aggregates and p-tau tangles are known to trigger NLRP3-mediated inflammasome activation, thus releasing IL-1β and other cytokines, and inducing proinflammatory microglia recruitment. And if that were not enough, hIAPP could aggravate all these events, even worsen the situation by the generation of crossseeding heterocomplexes of Aβ-hIAPP aggregates.

## Data Availability

Not applicable.

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
