# Peer review of "Insulin Resistance and Diabetes Mellitus in Alzheimer’s Disease"

_cells, 2021, doi:10.3390/cells10051236_

Round 1
Reviewer 1 Report
Major Concerns
Although it is a relevant topic, the review is excessively long and heavy. The authors describe the main enzymes and signaling pathways involved in insulin resistance in great detail in the first four sections. This description covers half of the review and is from section 5 (neurotransmitters), when the authors begin to correlate insulin resistance with different aspects of the central nervous system. It is from section 7 when the authors describe the pathophysiology of Alzheimer's disease and its correlation with T2DM. Thus, I strongly recommend that the authors consider significantly reducing the first four sections so that the review is more focused on the main topic of the manuscript.
The title provided by the editorial office does not match with the title in the manuscript (pdf). Please clarify.
Minor Concerns
- Check the Greek letters used throughout the manuscript, as some do not appear as such. For example, on page 2, line 67, "a" instead of alpha (a subunit); same in line 69.
- Figure 2 is not mentioned in the main text.
- Figure 3 is difficult to follow. I suggest splitting it or redesigned it for a better understanding, and the same with the legend of the figure.
Author Response
Reviewer 1
Major Concerns
- Although it is a relevant topic, the review is excessively long and heavy. The authors describe the main enzymes and signaling pathways involved in insulin resistance in great detail in the first four sections. This description covers half of the review and is from section 5 (neurotransmitters), when the authors begin to correlate insulin resistance with different aspects of the central nervous system. It is from section 7 when the authors describe the pathophysiology of Alzheimer's disease and its correlation with T2DM. Thus, I strongly recommend that the authors consider significantly reducing the first four sections so that the review is more focused on the main topic of the manuscript.
Thanks a lot for her/his comments about our manuscript. In this regard, we have significantly reduced the first four sections as suggested by this referee. Furthermore, we have revised the English language for improving the understanding.
- The title provided by the editorial office does not match with the title in the manuscript (pdf). Please clarify.
Thanks a lot for the comment. We have provided the correct title to the editorial office and corresponds to the title in the manuscript.
Minor Concerns
- Check the Greek letters used throughout the manuscript, as some do not appear as such. For example, on page 2, line 67, "a" instead of alpha (a subunit); same in line 69.
We have checked the Greek letters in the manuscript. The changes are highlighted in red
- Figure 2 is not mentioned in the main text.
Thanks a lot for the comment and we are sorry about this omission. Now, in the revised version of the manuscript we have included a mention to figure 2 (page 11 line 430 and 431 and highlighted in red)
- Figure 3 is difficult to follow. I suggest splitting it or redesigned it for a better understanding, and the same with the legend of the figure.
Thanks a lot for this comment. As suggested by the referee, we have separated figure 3 in the previous manuscript, into two different figures in the revised version of the review (new figure 3 and new figure 4).
Reviewer 2 Report
Insulin resistance and Diabetes Mellitus in Alzheimer´s disease
This paper is very informative and well written, and will be good guidance for the readers, hoping to understand the relationship between diabetes mellitus and Alzheimer´s disease (AD).
There are two concerns before the acceptance.
- In the Introduction, there is no description about AD. As you can see, the title has the word, Alzheimer´s disease. Please include summary description on AD, shown in the later sections.
2. Figure 3 is a bit complex to understand the important points showing critical events causing the cell and molecular level phenotypes of AD (for example, neuronal cell death, synaptic dysfunction, and so on).
Author Response
Reviewer 2
This paper is very informative and well written, and will be good guidance for the readers, hoping to understand the relationship between diabetes mellitus and Alzheimer´s disease (AD).
Thanks a lot to this referee for her/his comments about our manuscript. We are profoundly honored about it.
There are two concerns before the acceptance.
- In the Introduction, there is no description about AD. As you can see, the title has the word, Alzheimer´s disease. Please include summary description on AD, shown in the later sections.
Thanks a lot for your comments. We have included a summary description on Alzheimer´s disease in the introduction section and highlighted in red (from page 2 line 77 to page 3 line 83)
- Figure 3 is a bit complex to understand the important points showing critical events causing the cell and molecular level phenotypes of AD (for example, neuronal cell death, synaptic dysfunction, and so on).
Thanks a lot for the comment. Then, in order to make easier the understanding of the figure and suggested by the referee, we have splitted figure 3 in the previous manuscript, into two different figures in the revised version of the review (new figure 3 and new figure 4).
Round 2
Reviewer 1 Report
The authors have adequately addressed the points I raised. I have no further comments.